# Effects of Chitosan Oligosaccharide on Production Performance, Egg Quality and Ovarian Function in Laying Hens with Fatty Liver Syndrome

**DOI:** 10.3390/ani12182465

**Published:** 2022-09-18

**Authors:** Wenjing Tao, Feng Jin, Qiwen Fan, Na Zhao, Geng Wang, Encun Du, Fang Chen, Wanzheng Guo, Shaowen Huang, Mingxin Chen, Jintao Wei

**Affiliations:** 1Hubei Key Laboratory of Animal Embryo and Molecular Breeding, Institute of Animal Husbandry and Veterinary, Hubei Academy of Agricultural Sciences, Wuhan 430064, China; 2Key Laboratory of Molecular Animal Nutrition, College of Animal Science, Zhejiang University, Ministry of Education, Hangzhou 310058, China

**Keywords:** chitosan oligosaccharide, production performance, egg quality, ovarian function, laying hen

## Abstract

**Simple Summary:**

Fatty liver syndrome (FLS) often occurs in caged laying hens and can cause decreases in production performance. Chitosan oligosaccharide (COS), degraded from chitin or chitosan, has been demonstrated to prevent metabolic diseases in rodents. In this work, we found that dietary COS supplementation could improve production performance and egg quality in laying hens with FLS. Further study indicated that improved ovarian morphology and function may be involved in these beneficial effects of COS. Specifically, dietary COS supplementation decreased oxidative stress, inflammation and apoptosis in the ovaries of laying hens with FLS. This study provides evidence for the application of COS to improve production performance and egg quality in laying hens with FLS.

**Abstract:**

This study aimed to investigate the role of chitosan oligosaccharide (COS) as an additive in the feed of laying hens with fatty liver syndrome (FLS). Effects on production performance, egg quality as well as ovarian function were determined. A total of 360 Lohmann Pink-shell laying hens (28 weeks old) were randomly assigned to 5 groups (6 replicates × 12 birds). Hens were fed with a basal diet and a high-energy low-protein (HELP) diet supplemented with 0, 200, 400 and 800 mg/kg COS. COS reversed the lowered laying rates, increased feed-to-egg ratios and decreased albumen heights and Haugh units induced by the HELP diet. Additionally, COS improved the ovarian morphologies damaged by the HELP diet. Furthermore, COS enhanced antioxidant enzyme activities, reduced malonaldehyde levels and downregulated the mRNA expressions of nuclear factor kappa B, pro-inflammation cytokine genes and pro-apoptosis-related genes, while it upregulated the mRNA expression of anti-apoptosis-related genes in the ovaries of HELP-diet-fed hens. These findings suggested that dietary COS supplementation could improve production performance and egg quality in laying hens with FLS, and these beneficial effects were linked to improved ovarian morphology, which was attributed to decreased oxidative stress, inflammation and apoptosis in the ovaries.

## 1. Introduction

Fatty liver syndrome (FLS), characterized by excess lipid accumulation and various degrees of hemorrhage in the liver, often occurs in caged laying hens [1]. FLS can cause a sharp drop in the rate of egg production and can even induce the occurrence of sudden death, resulting in serious economic losses to the poultry industry [2]. There are multiple factors, including heredity, nutrition, hormones and environment, involved in the development of FLS, and nutrition is the leading cause of FLS [3]. In order to elucidate the pathogenesis of FLS and develop effective prevention and treatment strategies, a high-energy low-protein (HELP) diet is widely used to induce FLS in laying hens [4]. Previous studies have demonstrated that HELP diets may induce significant decreased egg production in laying hens [4,5]. The ovary is the key organ affecting reproductive capacity in females; thus, improving ovarian function is beneficial for the production performance of laying hens. It was reported that hepatic fat accumulation is closely associated with abnormal morphology and increased apoptosis in the ovaries of broiler breeder hens [6]. Moreover, a previous study showed that oxidative stress and inflammation in the ovaries are involved in the occurrence and development of FLS in laying hens [7]. These findings suggest that ovarian dysfunction may be the main contributing factor with respect to decreased egg production in laying hens with FLS.

Chitosan oligosaccharide (COS), degraded from chitin or chitosan, possesses various biological properties, such as antioxidant, anti-inflammatory, anti-obesity, hypolipidemic and hypoglycemic activities [8,9]. COS has been used to prevent metabolic diseases, including obesity, dyslipidemia, diabetes mellitus and hypertension [10,11,12,13]. COS could also prevent fatty liver disease in high-fat-diet-fed mice and rats, and these effects were linked to its antioxidant and anti-inflammatory activities [11,14,15]. In addition, COS has potential applications as a feed additive to promote the reproductive performance of livestock and poultry [16,17,18,19]. The beneficial effects of COS on reproductive performance were linked to improved ovarian function in sows [20]. When used as an alternative to antibiotics for laying hens, COS could improve production performance and egg quality under normal physiology conditions [18,19]. Moreover, COS could attenuate intestinal oxidative stress and inflammation in laying hens [21]. COS could also alleviate oxidative stress and inflammation in the livers, spleens and intestines of broilers [22,23]. However, until now, no study has demonstrated the effect of COS on production performance and egg quality in laying hens with FLS. Whether COS could suppress ovarian oxidative stress, inflammation and apoptosis in laying hens with FLS has not been clarified. Therefore, the purpose of the present study was to evaluate the effects of dietary COS supplementation on production performance, egg quality and ovarian function in laying hens with FLS.

## 2. Materials and Methods

### 2.1. Animals, Diets and Experimental Design

The experimental protocol was approved by the Animal Care and Use Committee of Hubei Academy of Agricultural Sciences (HBAAS20210007).

A total of 360 Lohmann Pink-shell laying hens (28 weeks old) were randomly assigned to 5 dietary treatments as follows: normal control (NC) group, HELP group, HELP + low dosage of COS (HELPLC) group, HELP + medium dosage of COS (HELPMC) group, HELP + high dosage of COS (HELPHC) group. Each treatment had 6 replicates with 12 hens per replicate. Birds in the NC group were fed a basal diet, and birds in the HELP, HELPLC, HELPMC and HELPHC groups received a HELP diet supplemented with 0, 200, 400, and 800 mg/kg COS, respectively. The basal diet was formulated according to the requirement of laying hens [24], and the HELP diet composition was based on our preliminary experimental results. The compositions and nutrient levels of the basal and HELP diets are shown in Table 1. The COS was obtained from Zhongkerongxin Biotechnology Co., Ltd. (Suzhou, China), with a purity of approximately 90%. The doses of COS were selected based on a previous study [19]. The hens were housed in 3-tier ladder-type cages (6 birds per cage, 80 × 80 × 50 cm) in an environmentally controlled room with a 16L:8D photoperiod. The diets and water were provided ad libitum. The experiment lasted for 14 weeks, including a 2-week adaptation period and a 12-week experimental period.

### 2.2. Production Performance and Sample Collection

The number and weight of qualified eggs were recorded daily for each replicate. Feed consumption was recorded weekly, and mortality was recorded daily throughout the experiment. The laying rate, daily feed intake and feed-to-egg ratio were calculated for each trial phase and for the overall experimental period. At the end of the trial, one hen from each replicate was randomly selected and sacrificed by jugular exsanguination. The ovary samples were collected for further determination.

### 2.3. Egg Quality

At the end of the trial, egg samples (8 eggs per replicate, 48 eggs per treatment) were obtained to analyze egg quality. All eggs were weighed using an electronic scale. The longitudinal and transverse diameters of the eggs, as well as the eggshell thicknesses, were measured using a vernier caliper. The egg shape index was calculated as egg longitudinal diameter divided by egg transverse diameter, and eggshell thickness was calculated by averaging three thickness values form the blunt end, middle and sharp end of the shell. Eggshell strength was determined with an eggshell strength meter (NFN388, Fujihira Industry Co. Ltd., Tokyo, Japan). Yolk color was evaluated by comparison with a color fan (Robotmation Co., Ltd., Tokyo, Japan), and the yolk weight ratio was calculated as yolk weight divided by egg weight. Albumen heights and Haugh units were measured using an Egg Multi Tester (EMT-7300, Robotmation, Japan).

### 2.4. Histopathological Examination

The ovary samples were washed with normal saline and then fixed in 4% paraformaldehyde solution. After fixing for 24 h, the tissues were embedded in paraffin. Then, the tissues were cut into slices of 5 µm thickness and stained with hematoxylin and eosin (H&E). The samples were observed and photographed using a light microscope.

### 2.5. Determination of Ovarian Antioxidant Capacity

The ovary samples were homogenized in ice-cold 0.9% saline at a ratio of 1:9 (wt/vol). Then, the supernatants were transferred into new Eppendorf tubes after centrifugation at 3000 rpm for 10 min. The activities of total antioxidant capacity (T-AOC), glutathione peroxidase (GSH-Px) and superoxide dismutase (SOD) and the levels of malondialdehyde (MDA) in the supernatants of the ovaries were assessed using commercial kits from the Nanjing Jiancheng Bioengineering Institute (Nanjing, China), according to the manufacturer’s instructions.

### 2.6. Quantitative Real-Time Polymerase Chain Reaction (PCR) Analysis

Total RNA was extracted from the ovary samples using TRIzol Reagent (TaKaRa, Dalian, China), and cDNAs were synthesized from 500 ng of RNA using a reverse transcription kit (TaKaRa, Dalian, China), according to the manufacturer’s instructions. Quantitative real-time PCR was performed on a LightCycler 96 PCR System (Roche, Mannheim, Germany) with TB Green Premix Ex Taq II (TaKaRa, Dalian, China). The sequences of primers used for PCR amplification are listed in Table 2. The mRNA expression levels of the target genes were normalized to the housekeeping gene glyceraldehyde-3-phosphate dehydrogenase (*GAPDH*) and calculated using the 2^−ΔΔ*C*T^ method.

### 2.7. Statistical Analysis

All data were analyzed by one-way ANOVA followed by Duncan’s test for multiple comparisons using SPSS 20.0 software (SPSS, Chicago, IL, USA). Differences among means were considered significant at levels of *p* < 0.05.

## 3. Results

### 3.1. Production Performance

The effects of dietary COS supplementation on the production performance of the laying hens are presented in Table 3. There were no significant differences in daily feed intake among all the groups during the trial (*p* > 0.05). From the 1st to the 4th week, the laying rates of hens in the HELP group were lower than those of hens in the NC group, whereas dietary supplementation with COS (400 or 800 mg/kg) obviously increased the laying rates (*p* < 0.05). Additionally, from the 1st to the 4th week, no difference was observed in feed-to-egg ratios among all the groups (*p* > 0.05). From the 5th to the 8th week, the hens in the HELP group showed significantly lower laying rates and higher feed-to-egg ratios than those in the NC group (*p* < 0.05), whereas dietary supplementation with COS (200, 400 or 800 mg/kg) obviously increased the laying rates and decreased the feed-to-egg ratios (*p* < 0.05). From the 9th to 12th week, the hens in the HELP group showed significantly lower laying rates and higher feed-to-egg ratios than those in the NC group (*p* < 0.05), whereas, compared with hens in the HELP group, laying rates increased and the feed-to-egg ratios decreased in hens in the HELPMC and HELPHC groups (*p* < 0.05). From the 1st to the 12th week, the hens in the HELP group showed significantly lower laying rates and higher feed-to-egg ratios than those in the NC group (*p* < 0.05), whereas, compared with hens in the HELP group, laying rates were increased in hens in the HELPLC, HELPMC and HELPHC groups (*p* < 0.05), and feed-to-egg ratios were decreased in the hens in the HELPMC and HELPHC groups (*p* < 0.05).

### 3.2. Egg Quality

The effects of dietary COS supplementation on the egg quality of the laying hens are presented in Table 4. There were no significant differences in egg weight, egg shape index, yolk color, yolk weight ratio, eggshell strength or eggshell thickness among all the groups (*p* > 0.05). The hens in the HELP group showed significantly lower albumen height and Haugh unit values than those in the NC group (*p* < 0.05), whereas, compared with hens in the HELP group, albumen heights and Haugh units were increased in the hens in the HELPMC and HELPHC groups (*p* < 0.05).

### 3.3. Histopathology

The effects of COS supplementation on ovarian morphology in the laying hens are presented in Figure 1. The morphological structures of the ovaries in the hens in the NC group were normal, with orderly arrangements of granulosa cells. The ovaries in the hens in the HELP group showed decreased numbers of granulosa cells and loose and irregular arrangements of granulosa cells, whereas dietary COS supplementation, especially at the dosages of 400 or 800 mg/kg, inhibited ovary damage by alleviating these histological alterations.

### 3.4. Antioxidant Capacities of the Ovaries

As shown in Figure 2, there were no significant differences in ovarian GSH-Px activity among all the groups (*p* > 0.05). The hens in the HELP group showed significantly lower activities of ovarian T-AOC and higher levels of MDA compared with those in the NC group (*p* < 0.05), and there were no significant differences in ovarian SOD activities between the HELP group and the NC group (*p* > 0.05). When compared with hens in the HELP group, ovarian T-AOC and SOD activities were significantly increased in the hens in the HELPMC and HELPHC groups (*p* < 0.05), and MDA levels were significantly decreased in the hens in the HELPLC, HELPMC and HELPHC groups (*p* < 0.05).

### 3.5. The mRNA Expression of Inflammation-Related Genes in the Ovaries

As shown in Figure 3, mRNA expression levels of ovarian nuclear factor kappa B (*NF-κB*), tumor necrosis factor-α (*TNF-α*), interleukin-1β (*IL-1β*) and interleukin-6 (*IL-6*) in the hens in the HELP group were significantly upregulated compared with those in the hens in the NC group (*p* < 0.05). Dietary supplementation with COS (400 or 800 mg/kg) obviously downregulated ovarian *NF-κB* mRNA expression (*p* < 0.05), and the COS administration (200, 400 or 800 mg/kg) significantly decreased the mRNA expression levels of *TNF-α*, *IL-1β* and *IL-6* in the ovaries (*p* < 0.05).

### 3.6. The mRNA Expression of Apoptosis-Related Genes in the Ovaries

As shown in Figure 4, there were no significant differences in ovarian pro-apoptosis-related gene *caspase 3* mRNA expression among all the groups (*p* > 0.05). The mRNA expressions of ovarian pro-apoptosis-related genes, including *caspase 8*, *caspase 9* and B cell lymphoma-2 associated X protein (*Bax*), were significantly higher in the HELP group than in the NC group (*p* < 0.05), whereas dietary supplementation with COS (200, 400 or 800 mg/kg) obviously downregulated the mRNA expressions of *caspase 8* and *caspase 9* in the ovaries (*p* < 0.05); however, ovarian *Bax* mRNA expression was not affected by dietary COS supplementation (*p* > 0.05). Additionally, no difference was observed in ovarian anti-apoptosis-related gene B cell lymphoma-2 (*Bcl-2*) mRNA expression between the HELP group and the NC group (*p* > 0.05), while the hens administrated with COS (400 or 800 mg/kg) showed significantly higher *Bcl-2* expression compared with the hens in the HELP group (*p* < 0.05).

## 4. Discussion

### 4.1. Production Performance

Laying hens with FLS often experience decreases in reproductive performance [2]. In this study, a HELP diet was used to induce FLS in laying hens, which is a typical approach taken to establish an FLS model [4]. We found that hens fed with a HELP diet showed lower laying rates and higher feed-to-egg ratios, which were consistent with the results of previous studies [4,7]. It has been reported that COS could improve production performance in laying hens under normal physiological conditions [18,19]. Our study is the first to demonstrate the beneficial effects of COS on laying rates and feed-to-egg ratios in laying hens with FLS. It is generally accepted that feed intake will decrease along with increase in dietary energy in laying hens [25]. Surprisingly, we found there was no significant difference in daily feed intake between the HELP group and the NC group. This result was inconsistent with those of previous studies [4,5], which indicated that HELP-diet-fed hens experienced decreased feed intake. However, Wu et al. reported that no difference was found in feed intake between the control group and an FLS group (induced by a high-fat diet) of laying hens [26]. Actually, according to the previous studies, dietary energy levels showed inconsistent effects on feed intake in laying hens. Zhang and Kim reported that the feed intake of hens fed a diet with 2800 kcal ME/kg was decreased when compared to those fed a diet with 2700 kcal ME/kg [27]. In other studies, changes in dietary energy levels did not affect feed intake in laying hens [28,29]. These discrepancies may be due to differences in genotype, age, diet composition or environment [30]. Additionally, dietary COS supplementation did not affect feed intake in laying hens with FLS, which was in agreement with the effect of COS supplementation in laying hens under normal physiological conditions [18,21].

### 4.2. Egg Quality

The literature on the effect of FLS on egg quality in laying hens is limited. Rozenboim et al. reported that egg weight was lower in a HELP group compared to the control [4]. In the present study, egg weight, egg shape index, yolk color, yolk weight ratio, eggshell strength and eggshell thickness were not affected by the HELP diet or COS supplementation. However, albumen heights and Haugh unit values were lower in the HELP group compared to the control, whereas dietary COS supplementation significantly increased albumen heights and Haugh units. To our knowledge, this is the first time the beneficial effect of COS on egg quality in laying hens with FLS has been reported. Similarly, previous studies on egg quality in laying hens under normal physiological conditions found that COS showed beneficial effects on egg-quality characteristics, including yolk color, eggshell strength, eggshell thickness, albumen height and Haugh unit [18,19,21,31,32]. Although the results were not entirely consistent, it can be concluded that COS is effective in improving egg quality in laying hens with or without FLS.

### 4.3. Ovarian Morphology

Metabolic disorders, including fatty liver disease, may cause abnormalities in the reproductive functions of humans and animals [33,34]. The ovary is the main organ in the reproductive system, and its dysfunction may be the major contributing factor with respect to decreased egg production and egg quality in laying hens with FLS. To explore the mechanisms underlying the effects of COS on production performance and egg quality in laying hens with FLS, ovarian morphology, ovarian antioxidant capacity and mRNA expression of genes related to ovarian inflammation and apoptosis were further examined. Walzem et al. reported that FLS induced follicular collapse and oviduct involution in laying hens [35]. Chen et al. found that hepatic fat accumulation is positively correlated with abnormal ovarian morphology in broiler breeder hens [6]. In our study, the histological examinations demonstrated that the numbers of granulosa cells were reduced and that the arrangements of the granulosa cells were damaged by the HELP diet in the laying hens, whereas dietary COS supplementation improved ovarian morphology.

### 4.4. Ovarian Antioxidant Capacity

Oxidative stress could induce ovarian dysfunction and result in reproductive failure [36]. Growing evidence has demonstrated that antioxidants may improve laying performance and ovarian function in hens [37,38]. In the present study, HELP-diet-fed hens showed lowered activities of antioxidant enzyme T-AOC and increased levels of the lipid peroxidation product MDA in the ovary, indicating that the ovaries of HELP-diet-fed hens suffered from increased oxidative stress and had decreased antioxidant capacities, which were consistent with the results of the study by Xing et al. [7]. Numerous studies in vitro and in vivo have demonstrated that COS possesses antioxidative properties [8]. Tao et al. reported that COS reduced hepatic oxidative stress in mice with fatty liver disease [14]. Yang et al. showed that COS alleviated hydrogen peroxide-stimulated oxidative damage in a human ovarian granulosa cell line [39]. Consistent with these findings, we found that dietary COS administration enhanced ovarian antioxidant capacity by increasing the activities of T-AOC and SOD and reducing levels of MDA.

### 4.5. Ovarian Inflammation

Fatty liver disease is closely associated with chronic systemic and local inflammation [40]. In women with polycystic ovary syndrome, a chronic ovarian pro-inflammatory state resulted in ovarian dysfunction [41]. In laying hens, the development of polycystic ovaries with cancer was related to chronic inflammation in the ovary [42]. There is only one study in the literature on the effect of FLS on ovarian inflammation in laying hens, reporting that inflammatory response was enhanced in the ovaries of hens with FLS [7]. Similarly, we found that HELP-diet-fed hens showed upregulated mRNA expression levels of *NF-κB*, *TNF-α*, *IL-1β* and *IL-6* in the ovaries. As a transcription factor, NF-κB plays a critical role in various inflammatory-associated metabolic diseases, and its activation induces the production of proinflammatory cytokines *TNF-α*, *IL-1β* and *IL-6* [43,44]. Due to its anti-inflammatory activity, COS has been used to prevent inflammatory-associated metabolic diseases [8,14]. In this study, dietary COS supplementation obviously downregulated mRNA expression of these inflammation-related genes, suggesting that COS inhibited ovarian inflammation in laying hens with FLS.

### 4.6. Ovarian Apoptosis

A previous study reported that hepatic fat accumulation is positively correlated with apoptosis in the ovaries in broiler breeder hens [6]. Additionally, it has been demonstrated that breeders with lower egg laying rates had higher ovarian-cell apoptosis rates [45]. Bcl-2 family proteins play important roles in the regulation of cellular apoptosis, among which Bcl-2 is the most representative anti-apoptotic protein, whereas Bax exhibits pro-apoptotic activity [46]. Upon receipt of an apoptotic signal, apoptotic initiators, including caspase 8 and caspase 9, are activated, and then a downstream apoptotic executor, such as caspase 3, is activated, ultimately resulting in apoptosis [47]. In the current study, we found that the mRNA expression levels of pro-apoptosis-related genes (*caspase 8*, *caspase 9* and *Bax*) in the ovaries of HELP-diet-fed hens were significantly higher than those of NC-diet-fed hens. Increased ovarian apoptosis may be the main contributing factor with respect to the reduced numbers of granulosa cells in hens with FLS. Yang et al. showed that COS inhibited apoptosis in a human ovarian granulosa cell line under oxidative stress induced by hydrogen peroxide [39]. In agreement with these results, we found that dietary COS supplementation downregulated the mRNA expression of *caspase 8* and *caspase 9* and upregulated the mRNA expression of the anti-apoptosis-related gene *Bcl-2* in the ovaries of HELP-diet-fed hens, suggesting that COS suppressed ovarian apoptosis in laying hens with FLS.

## 5. Conclusions

In conclusion, our results suggested that dietary COS supplementation, especially at the dosages of 400 or 800 mg/kg, could improve production performance and egg quality in laying hens with FLS, and these beneficial effects were linked to improved ovarian morphology, which was attributed to decreased oxidative stress, inflammation and apoptosis in the ovaries.

## Figures and Tables

**Figure 1 animals-12-02465-f001:**
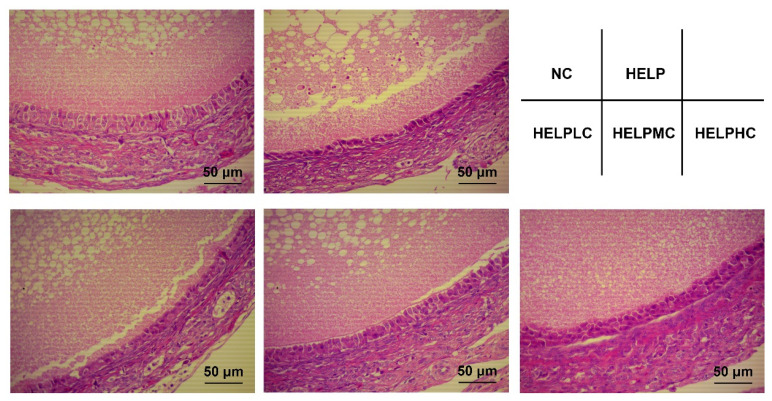
Effects of chitosan oligosaccharide on ovarian morphology in the laying hens. Representative images of the ovaries stained with hematoxylin and eosin. Abbreviations: HELP, high-energy low-protein; HELPLC, HELPMC and HELPHC, high-energy low-protein diet supplemented with 200, 400 and 800 mg/kg chitosan oligosaccharide, respectively; NC, normal control.

**Figure 2 animals-12-02465-f002:**
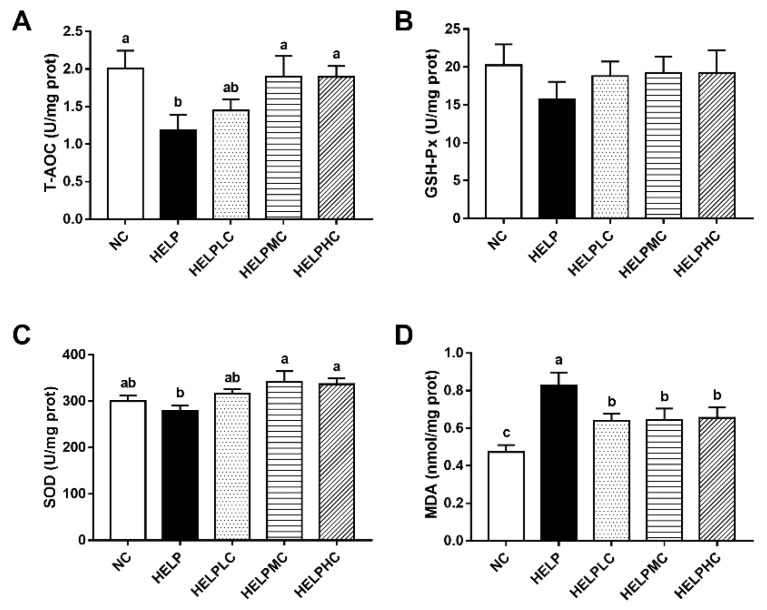
Effects of chitosan oligosaccharide on ovarian antioxidant capacity in the laying hens. (**A**) T-AOC, (**B**) GSH-Px, (**C**) SOD activities and (**D**) MDA levels in the ovaries. Each mean represents one layer/replicate, six replicates/treatment. Means lacking a common superscript are significantly different (*p* < 0.05). Abbreviations: GSH-Px, glutathione peroxidase; HELP, high-energy low-protein; HELPLC, HELPMC and HELPHC, high-energy low-protein diet supplemented with 200, 400 and 800 mg/kg chitosan oligosaccharide, respectively; MDA, malondialdehyde; NC, normal control; SOD, superoxide dismutase; T-AOC, total antioxidant capacity.

**Figure 3 animals-12-02465-f003:**
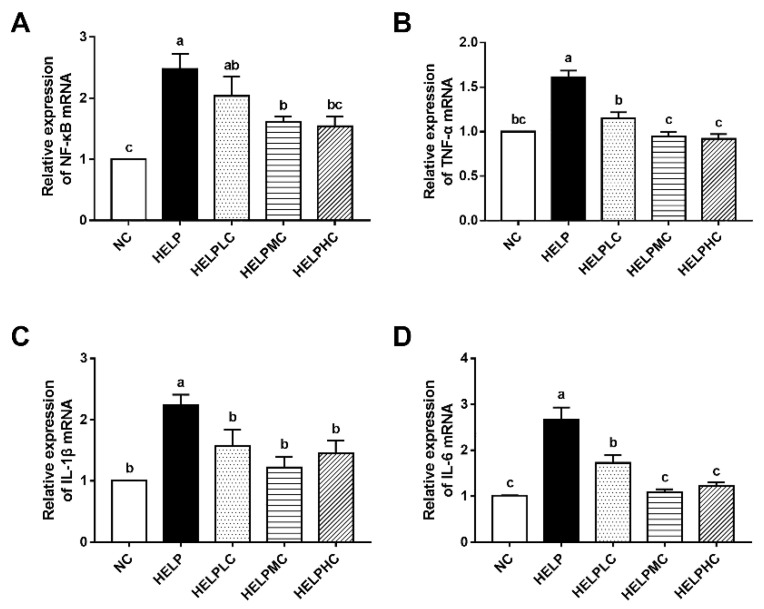
Effects of chitosan oligosaccharide on ovarian inflammation in the laying hens. Relative mRNA expression of (**A**) *NF-κB*, (**B**) *TNF-α*, (**C**) *IL-1β* and (**D**) *IL-6* in the ovaries. Each mean represents one layer/replicate, six replicates/treatment. Means lacking a common superscript are significantly different (*p* < 0.05). Abbreviations: HELP, high-energy low-protein; HELPLC, HELPMC and HELPHC, high-energy low-protein diet supplemented with 200, 400 and 800 mg/kg chitosan oligosaccharide, respectively; *IL-1β*, interleukin-1β; *IL-6*, interleukin-6; NC, normal control; *NF-κB*, nuclear factor kappa B; *TNF-α*, tumor necrosis factor-α.

**Figure 4 animals-12-02465-f004:**
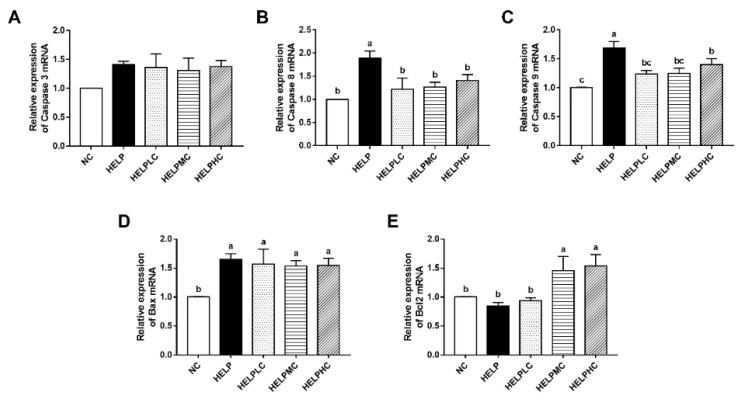
Effects of chitosan oligosaccharide on ovarian apoptosis-related gene expression in the laying hens. Relative mRNA expression of (**A**) *caspase 3*, (**B**) *caspase 8*, (**C**) *caspase 9*, (**D**) *Bax* and (**E**) *Bcl-2* in the ovaries. Each mean represents one layer/replicate, six replicates/treatment. Means lacking a common superscript are significantly different (*p* < 0.05). Abbreviations: *Bax*, B cell lymphoma-2 associated X protein; *Bcl-2*, B cell lymphoma-2; HELP, high-energy low-protein; HELPLC, HELPMC and HELPHC, high-energy low-protein diet supplemented with 200, 400 and 800 mg/kg chitosan oligosaccharide, respectively; NC, normal control.

**Table 1 animals-12-02465-t001:** Composition and nutrient levels of diets (air-dried basis).

Items (%, Unless Otherwise Indicated)	Basal Diet	High-Energy Low-Protein Diet
Ingredients		
Corn	63.00	66.00
Soybean meal	24.00	17.20
Limestone	8.00	8.00
Soybean oil	0.00	3.80
Premix ^1^	5.00	5.00
Total	100.00	100.00
Nutrient levels ^2^		
Metabolic energy (kcal/kg)	2602.20	2854.34
Crude protein	15.65	12.88
Lysine	0.79	0.62
Methionine	0.35	0.32
Calcium	3.41	3.39
Total phosphorus	0.43	0.40

^1^ Provided per kilogram of diet: Cu, 90 mg; Fe, 560 mg; Zn, 950 mg; Mn, 1080 mg; Se, 4.3 mg; I, 6 mg; vitamin A, 132,000 IU; vitamin D_3_, 45,000 IU; vitamin E, 280 mg; vitamin K_3_, 35 mg; vitamin B_1_, 18 mg; vitamin B_2_, 70 mg; vitamin B_6_, 36 mg; pantothenic acid, 110 mg; niacinamide, 285 mg; folic acid, 16 mg; biotin, 1.5 mg. ^2^ Calculated according to the Feed Database in China (2020).

**Table 2 animals-12-02465-t002:** Primer sequences used in this study.

Gene	Primer Sequences (5′–3′)	Size (bp)
*GAPDH*	F: TGCTGCCCAGAACATCATCC R: ACGGCAGGTCAGGTCAACAA	142
*NF-κB*	F: TCAACGCAGGACCTAAAGACAT R: GCAGATAGCCAAGTTCAGGATG	162
*TNF-α*	F: GCCCTTCCTGTAACCAGATG R: ACACGACAGCCAAGTCAACG	71
*IL-1β*	F: ACTGGGCATCAAGGGCTA R: GGTAGAAGATGAAGCGGGTC	131
*IL-6*	F: AGGACGAGATGTGCAAGAAGT R: TTGGGCAGGTTGAGGTTGTT	79
*caspase 3*	F: AAAGATGGACCACGCTCAGG R: TGAACGAGATGACAGTCCGG	204
*caspase 8*	F: AGTGAACAACTATCGGTGCAT R: CTTCCTGCCCATCAACACCA	107
*caspase 9*	F: TATGGTGGAGGACATGCAGA R: AATATTGGGAAGGCCTGCTT	99
*Bax*	F: GTACGTCAATGTGGTCACCC R: TGGGATAATGCTGGGGTTGA	210
*Bcl-2*	F: ACCATGAATGAAACCGTGCC R: TTGTCGTAGCCTCTTCTCCC	181

Abbreviations: *Bax*, B cell lymphoma-2 associated X protein; *Bcl-2*, B cell lymphoma-2; bp, base pair; F, forward; *IL-1β*, interleukin-1β; *IL-6*, interleukin-6; *NF-κB*, nuclear factor kappa B; R, reverse; *TNF-α*, tumor necrosis factor-α.

**Table 3 animals-12-02465-t003:** Effects of chitosan oligosaccharide on production performance in the laying hens.

Items ^1^	NC	HELP	HELPLC	HELPMC	HELPHC	SEM	*p*-Values
1st to 4th week							
Laying rate (%)	94.79 ^a^	92.07 ^b^	94.21 ^ab^	95.66 ^a^	95.54 ^a^	0.430	0.042
Daily feed intake (g/hen/d)	111.45	111.49	111.53	111.68	111.56	0.041	0.461
Feed-to-egg ratio	2.08	2.15	2.12	2.06	2.07	0.011	0.054
5th to 8th week							
Laying rate (%)	92.88 ^a^	86.34 ^b^	90.80 ^a^	92.77 ^a^	92.48 ^a^	0.620	<0.001
Daily feed intake (g/hen/d)	111.07	111.05	111.14	111.16	111.16	0.023	0.416
Feed-to-egg ratio	2.10 ^b^	2.27 ^a^	2.16 ^b^	2.10 ^b^	2.11 ^b^	0.017	0.002
9th to 12th week							
Laying rate (%)	90.97 ^a^	83.00 ^b^	86.34 ^ab^	88.08 ^a^	89.06 ^a^	0.795	0.011
Daily feed intake (g/hen/d)	107.63	108.27	107.24	106.99	106.98	0.233	0.379
Feed-to-egg ratio	2.11 ^b^	2.37 ^a^	2.25 ^ab^	2.20 ^b^	2.15 ^b^	0.027	0.016
1st to 12th week							
Laying rate (%)	92.88 ^a^	87.14 ^b^	90.45 ^a^	92.17 ^a^	92.36 ^a^	0.568	0.002
Daily feed intake (g/hen/d)	110.05	110.27	109.97	109.94	109.90	0.080	0.650
Feed-to-egg ratio	2.10 ^b^	2.26 ^a^	2.18 ^ab^	2.12 ^b^	2.11 ^b^	0.018	0.012

Abbreviations: HELP, high-energy low-protein; HELPLC, HELPMC and HELPHC, high-energy low-protein diet supplemented with 200, 400 and 800 mg/kg chitosan oligosaccharide, respectively; NC, normal control. ^a,b^ Means within a row lacking a common superscript differ significantly (*p* < 0.05). ^1^ Each mean represents 12 layers/replicate, 6 replicates/treatment.

**Table 4 animals-12-02465-t004:** Effects of chitosan oligosaccharide on egg quality in the laying hens.

Items ^1^	NC	HELP	HELPLC	HELPMC	HELPHC	SEM	*p*-Values
Egg weight (g)	55.74	55.84	55.62	55.31	55.72	0.211	0.955
Egg shape index	1.33	1.33	1.33	1.33	1.34	0.003	0.844
Yolk color	6.21	6.63	6.33	6.48	6.25	0.075	0.376
Yolk weight ratio (%)	26.66	26.91	27.47	26.92	26.99	0.134	0.447
Eggshell strength (N)	44.98	48.32	47.74	47.58	48.42	0.557	0.288
Eggshell thickness (μm)	363.45	361.85	360.83	363.33	363.47	1.465	0.975
Albumen height (mm)	6.89 ^ab^	6.44 ^c^	6.65 ^bc^	6.88 ^ab^	7.15 ^a^	0.070	0.008
Haugh unit	84.02 ^ab^	80.79 ^c^	82.45 ^bc^	84.05 ^ab^	85.58 ^a^	0.445	0.003

Abbreviations: HELP, high-energy low-protein; HELPLC, HELPMC and HELPHC, high-energylow-protein diet supplemented with 200, 400 and 800 mg/kg chitosan oligosaccharide, respectively; NC, normal control. ^a–c^ Means within a row lacking a common superscript differ significantly (*p* < 0.05). ^1^ Each mean represents eight eggs/replicate, six replicates/treatment.

## Data Availability

The data presented in this study are available upon request from the corresponding author.

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
