# Peer review of "Effects of Chitosan Oligosaccharide on Production Performance, Egg Quality and Ovarian Function in Laying Hens with Fatty Liver Syndrome"

_animals, 2022, doi:10.3390/ani12182465_

Round 1
Reviewer 1 Report
The authors state that “the purpose of the present study was to evaluate the effects of dietary COS supplementation on production performance, egg quality and ovary function in laying hens with FLS”.
The mechanisms underlying hepatic steatosis include an oversupply of free fatty acids to the liver, interference with the triglyceride cycle, increases in the synthesis or esterification of fatty acids, decreased fatty acid oxidation, decreased apoprotein synthesis, and decreased synthesis or secretion of very low density lipoprotein.
Why didn't the authors determine the lipid and triglyceride contents in the liver? please clarify.
Line 61 change “activities [8,9]” as “activities [8,9,10,11]”.
10 Lan, R.; Wei, L.; Chang, Q., Wu, S.; Zhihui, Z. Effects of dietary chitosan oligosaccharides on oxidative stress and inflammation response in liver and spleen of yellow-feather broilers exposed to high ambient temperature. It. J. Anim. Sci. 2020, 19, 1508–1517. https://doi.org/10.1080/1828051X.2020.1850215
11 Cheng, Y.; Xie, Y.; Shi, L.; Xing, Y.; Guo, S.; Gao, Y.; Liu, Z.; Yan, S.; Shi, B. Effects of rare earth-chitosan chelate on growth performance, antioxidative and immune function in broilers. It. J. Anim. Sci. 2022, 21, 303-313. https://doi.org/10.1080/1828051X.2022.2028589
Reviewer 2 Report
The manuscript presented for the review presents the original results of studies that can make an important contribution to the current state of the art regarding the presented topic. In the reviewer's opinion, the Introduction indicates the state of current knowledge. The experimental design is appropriate to resolve the stated objectives of the study. The experimental techniques are appropriate to resolve the stated objectives of the study. The results are presented in an unbiased fashion. They are presented in a clear, concise and complete manner. The discussion is relevant and adequate for full interpretation of results. The results and discussion justify the conclusions drawn from the work. However, I propose to divide the Discussion chapter into similar subsections, which are located in the Results chapter. Moreover, the References lacks a DOI number or the address of a website where you can find the cited publication.
